# Recombinant *Potyvirus lilimaculae* in asymptomatic *Galanthus nivalis*: Ecological and evolutionary implications

János Ágoston[1], Asztéria Almási[2], Katalin Salánki[2], László Palkovics[1,3,4]*

1 HUN-REN-SZE PhatoPlant-Lab, Széchenyi István University, Mosonmagyaróvár, Hungary, 2 HUN-REN Centre for Agricultural Research, Plant Protection Institute, Budapest, Hungary, 3 Department of Plant Sciences, Albert Kázmér Faculty of Mosonmagyaróvár, Széchenyi István University, Mosonmagyaróvár, Hungary, 4 Agricultural and Food Research Centre, Albert Kázmér Faculty of Mosonmagyaróvár, Széchenyi István University, Mosonmagyaróvár, Hungary

* palkovics.laszlo.amand@sze.hu

## Abstract

Snowdrops are economically important early-spring flowering geophytes. They are protected both in Hungary and across the European Union, and their international trade is regulated under the Washington Convention. Despite their importance, virological research on *Galanthus* species and cultivars remains limited. To date, only a single virus, Snowdrop virus Y, a tentative member of the genus *Potyvirus* within the family *Potyviridae*, has been identified in *Galanthus*. In this study, ten wild leaf samples and one cultivated, asymptomatic leaf samples were collected in accordance with the prescribed permit conditions. Both ELISA and RT-PCR tests were conducted, indicating infection by a Potyvirus. Direct sequencing of the PCR products revealed nucleotide heterogeneity in one sample, suggesting infection with multiple isolates of the same virus. In this case, the purified PCR product was ligated into pGEM®-T Easy vector. Individual clones were then sequenced to identify the distinct viral isolates present. A BLAST analysis of the sequences revealed that all three snowdrop isolates shared at least 96% identity in the complete coat protein region with the *Potyvirus lilimaculae* (lilí mottle virus, LMoV) isolate Handan (JF714974), previously identified in *Narcissus tazetta* var. *chinensis* from Hubei Province, China. To confirm pathogenicity, sap inoculation was performed on ELISA- and RT-PCR-negative snowdrops. One year later, these plants were re-tested via the same methods, which confirmed successful infection. Phylogenetic analysis revealed that all snowdrop isolates clustered within subgroup II and presented strong recombination signals. The detection of LMoV in asymptomatic wild plants has major implications for nature conservation, horticulture, virus reservoir identification, phytosanitary regulation, and our understanding of *Potyviridae* evolution.

**Citation:** Ágoston J, Almási A, Salánki K, Palkovics L (2026) Recombinant *Potyvirus lilimaculae* in asymptomatic *Galanthus nivalis*: Ecological and evolutionary implications. PLoS One 21(3): e0345337. https://doi.org/10.1371/journal.pone.0345337

**Data availability statement:** All relevant data are within the manuscript.

**Funding:** HUN-REN Hungarian Research Network (project number: 3200107). JA's research was partly funded by the North American Lily Society Research Trust Fund Grant (grant number: 2024-02). The funders had no role in study design, data collection and analysis, decision to publish, or preparation of the manuscript.

**Competing interests:** The authors have declared that no competing interests exist.

## Introduction

Snowdrops (*Galanthus* spp.) are perennial, bulbous plants belonging to the family Amaryllidaceae [1–4]. The genus currently comprises 23 accepted species, native to Europe and the Middle East [5]. They are valued for their ornamental appeal and versatility [6]. Under moist, humus-rich soil conditions, they readily naturalize in garden settings through seed dispersal or bulb offsets [2–4,6–8]. Typically, bulbs are planted in early to late autumn in borders, rockeries, beneath trees, or lawns to encourage naturalization [1,6,7].

The most commonly cultivated species include *G. elwesii*, *G. nivalis*, *G. plicatus* and *G. woronowii* [3,6,9,10].

Currently 275 cultivars and forms of *Galanthus* are officially registered [11,12], although many additional unregistered variants exist. Rare or unusual clones are propagated only vegetatively and are highly sought after by specialist growers and collectors, who are often referred to as "galantophiles" [9,10]. These prized cultivars can command exceptionally high market prices because of their rarity and exclusivity.

Snowdrops are also recognized as important medicinal plants. Their bulbs are natural sources of pharmacologically active alkaloids such as galantamine and lycorine. These compounds exhibit acetylcholinesterase inhibitor activity and are used to treat Alzheimer's disease, certain cancers and other medical conditions [13–18]. However, owing to the presence of these alkaloids, all parts of the snowdrop plant are toxic to humans and most vertebrates [14].

Snowdrops have been extensively collected from their native habitats, leading to notable population declines across their range [1–3,7]. As a result, the entire *Galanthus* genus has been listed under Appendix II of the Convention on International Trade in Endangered Species of Wild Fauna and Flora (CITES) since January 18, 1990 [14]. Although the collection of snowdrop bulbs from the wild continues under special permit conditions – particularly in countries such as Turkey and Georgia – there is an increasing emphasis on sustainable production through nursery cultivation [19–22].

Since 2005, *Galanthus nivalis* has been listed in Hungary under Decree No. 13/2001, Appendix 7, titled " Plant species of importance for nature conservation in the European Community", assigning a monetary value of 10,000 HUF per individual plant [23]. According to a public awareness poster issued by the Nature Conservation Office of the Ministry of Environment and Water Management, over-collection threathens all native snowdrop populations in Hungary [24]. Nonetheless, snowdrops planted prior to this designation continue to be cultivated in large numbers in private gardens and public parks without permits.

Today, the vast majority of Hungary's *Galanthus* populations are located within protected or strictly protected Natura2000 sites, which are part of the country's National Park system [25]. Over-collection and poaching of bulbs have severely impacted the species' ability to regenerate. Under nursery conditions, it typically takes four to five years for a bulb to reach flowering size; in natural habitats, this process may take even longer and seed production is usually low compared to other genera like *Lilium* or *Muscari* [7].

Under natural conditions a snowdrop bulb typically produces a daughter bulb every 2–3 years. To accelerate vegetative propagation bulbs can be chipped (cut into sections), twin-scaled [7,26–28], or propagated through tissue culture (TC) [7,29–35].

Data on virus infections in *Galanthus* species and cultivars are lacking. The main reason for this knowledge gap is the difficulty of symptom detection, as most *Galanthus* varieties possess very glaucous leaves that obscure visible signs of infection. When symptoms do occur, they typically manifest as yellow-green stripes on the foliage [3]. However, symptom expression is inconsistent and may not appear every year, further complicating detection and diagnosis.

To date, only a single virus has been described in *Galanthus*: Snowdrop virus Y (SVY), a tentative member of the genus *Potyvirus* within the family *Potyviridae*. The first nucleotide sequence of Snowdrop virus Y was from *Galanthus* 'Gerald Parker' collected in 2008 from the UK, although it was not published in a peer-reviewed journal [36]. Subsequently, SVY was detected in Australia in a symptomatic leaf of *Muscari neglectum* via high-throughput sequencing [37]. In 2021, the virus was reported in *Narcissus* from South Korea and was later officially confirmed in *Narcissus* plants in the UK through ELISA, RT-PCR and Illumina sequencing [38].

The aim of this study was to investigate the presence of *Potyvirus* infections within wild and cultivated *Galanthus* taxa in Hungary.

## Materials and methods

Special permits were required for entering the protected sites, sample collection, importation of plants to Hungary, and growing the pants and carrying out the research, which were granted by permits: PE-KTFO/5659-17/2019, PE-06/ÉTKF/02444–6/2019 and PE-06/ÉTKF/02444–7/2019.

In accordance with the permit conditions, 200 mg of leaf tissue was collected from randomly selected plants at various sampling locations and dates. Where possible, samples were preferentially taken from plants growing in tufts, as these were presumed to be clones of a single seed-grown individual. This strategy was intended to minimize the ecological impact of sampling on both the individual plants and their surrounding environment. The detailed sample collection data are presented in Table 1.

The samples were shipped on ice to the laboratory, where they were cut into ~50 mg leaf discs, immediately frozen, and stored long-term at −70 °C.

Table 1. Data collected from asymptomatic snowdrop samples in Hungary.

| Sample ID | Taxon | Collection date | Nearest municipality | Protection type | Number of shoots in the sampled tuft |
|---|---|---|---|---|---|
| 647 | *Galanthus nivalis* | 2021-Mar-21 | Miskolc | Natura2000 | 11 |
| 659 | *Galanthus nivalis* subsp. *imperatii* | 2020-Feb-20 | Solt | Natura2000 | 1 |
| 666 | *Galanthus nivalis* subsp. *imperatii* | 2020-Feb-26 | Miskolc | Natura2000 | 8 |
| 685* | *Galanthus nivalis* subsp. *imperatii* | 2019-Apr-01 | Miskolc | none, home garden | 23 |
| 785 | *Galanthus nivalis* subsp. *imperatii* | 2020-Feb-20 | Kővágószőlős | Natura2000 | 21 |
| 804 | *Galanthus nivalis* | 2020-Apr-26 | Solt | Natura2000 | 13 |
| 857 | *Galanthus nivalis* | 2021-Mar-21 | Miskolc | Natura2000 | 9 |
| 862 | *Galanthus nivalis* | 2021-Mar-28 | Solt | Natura2000 | 5 |
| 863 | *Galanthus nivalis* | 2021-Mar-22 | Kővágószőlős | Natura2000 | 20 |
| 865 | *Galanthus nivalis* | 2021-Mar-21 | Miskolc | Natura2000 | 12 |
| 885 | *Galanthus nivalis* | 2020-Apr-28 | Gyöngyös | Natura2000 | 9 |

*: bulbs were planted before *G. nivalis* became a protected species in Hungary.

## Serological tests

Potyvirus group specific ELISA tests were performed using the monoclonal antibody PTY1 (Agdia Cat# 27200, RRID:AB_2819158; Agdia EMEA, Soisy-sur-Seine, France) raised in mice against the potyvirus coat protein. The enzyme conjugate was a polyclonal antibody against mouse IgG and was raised in rabbits. The final dilutions of both the detection antibody and the enzyme conjugate were 1:100. Tests were conducted in duplicate with one leaf disc per sample, following the manufacturer's instructions [39]. Positive control was supplied by the manufacturer, while the negative control was prepared from *Chenopodium foetidum* seedlings grown in a vector-free growth chamber. After the addition of 1 mg/ml *para*-Nitrophenylphosphate (pNPP) in substrate buffer, the ELISA plates were incubated in the dark at 37 °C. The absorbance was measured at 405 nm using an ELISA reader after 1 hour of incubation. Each optical density (OD) reading was corrected by subtracting the blank value. A sample was considered positive if its corrected OD value was at least three times greater than the corrected average OD of the negative controls.

## Molecular identification of virus infection

**Nucleic acid extraction, reverse transcription and PCR.** One leaf disc per sample was homogenized in an ice-cold mortar with approximately 20–25 mg of carborundum (500 mesh fraction). Total nucleic acids were extracted using a modified CTAB protocol as described by Xu *et al.* [40]. First-strand cDNA synthesis was performed using the Maxima H Minus First Strand cDNA Synthesis Kit (Thermo Fisher Scientific Baltics UAB, Vilnius, Lithuania) with random hexamer primers, following the manufacturer's instructions. The RT-PCR analyses were conducted to detect viruses from multiple genera using *Taq* DNA Polymerase, recombinant (Thermo Fisher Scientific Baltics UAB, Vilnius, Lithuania) according to the manufacturer's instructions. *Potyviruses* were targeted using the poty7941 and polyT$_2$ primers [41], the *Nepovirus* subgroup A was tested using Nepo-A forward and reverse primers [42] while *Orthotospovirus tomatomaculae* (Tomato spotted wilt virus, TSWV) was detected using the TSWV S forward and reverse primers [43], following previously published protocols.

Specific PCR products were excised from 1×TBE agarose gels and purified using the High Pure PCR Product Purification Kit (Roche, Mannheim, Germany). Nucleotide sequences of the PCR products or cloned inserts were determined via the Sanger dideoxy sequencing method [44], which was performed by commercial service providers. In the samples with heterogenous isolates, the purified PCR product was ligated into the pGEM®-T Easy vector (Promega, Wisconsin, USA) and transformed into *E. coli* DH5α cells. Individual clones were subsequently sequenced to identify all distinct viral isolates present. For molecular identification, the resulting sequences were analyzed via megaBLAST searches [45]. Species-level identification of *Potyvirus* isolates was performed according to the demarcation criteria established by the latest taxonomy profile of the International Committee ont he Taxonomy of Viruses (ICTV) [46].

## Sap inoculation

A sap-inoculation experiment was attempted using *G. nivalis* plants that had been acquired already growing and transplanted "in the green" from the Netherlands in late February and early March 2020. Unfortunately, these bulbs dried out during the COVID-19 lockdown and did not survive.

Seed propagation was attempted by sowing 47 wild-collected seeds in September 2020, in accordance with permit conditions and using the method described below. However, no germination was observed in 2021 or thereafter. This failure was likely due to seed desiccation or removal by ants, which may have transported the elaiosome-bearing seeds to their nests.

Commercially grown vegetatively propagated *G. elwesii* bulbs were used as test plants. Bulbs were individually potted in September 2020 in 9 cm square pots (Lamprecht-Verpackungen GmbH, Göttingen, Germany) to a depth of 3 cm. The potting medium consisted of 50% Klasmann TS4 substrate and 50% washed coir, supplemented with 3 g/l Osmocote Exact High K DCT 8-9M (12-7-19+TE) (Everris International B.V., Heerlen, Netherlands). Pots were placed outdoors in

the shade of an *Acer pseudoplatanus* tree. Irrigation with tap water (pH 7.8, EC 574 µS/cm) was provided as needed to keep the substrate consistently moist, as *Galanthus* bulbs are highly sensitive to desiccation [9,10,47].

In early January 2021, the potted bulbs were transferred to a vector-free, unheated greenhouse to induce sprouting. At flowering, all plants were screened using ELISA and RT-PCR for the viruses listed above. Only virus-negative plants were used in subsequent inoculation experiments.

Inoculations were performed at the late flowering stage. For each plant, 200 µl of 0.01 M Sörensen's phosphate buffer (pH 7.0) containing 20 mg of 500 mesh carborundum was used. Leaves were dusted with carborundum; mock-inoculated plants received only buffer, in the case of sap inoculated plants 1 leaf disc was homogenized in the buffer. A sterile glass rod was used to gently rub the leaf surface, facilitating micro-wounding and absorption of the inoculum. All the plants were re-tested in the following growing season using both ELISA and RT-PCR to assess successful infection.

## Phylogenetic analyses

Phylogenetic analyses were performed using MEGA X software [48], on the basis of the nucleotide (nt) sequences obtained from the virus coat protein (CP) region. The sequence alignments included the isolates characterized in this study, along with reference sequences obtained from NCBI GenBankwhich included lily mottle virus (EU348826, JN127341, JN848600, JQ361099, KF417755, MF781080, MF983709, MK368784, MK368788, MK368790, MK368792, MK368793, MK368794, MK368801, MK368802, MK368803, MK368804, MK368806, MK368809, OM311163) and snowdrop virus Y(EU927399, LC757029, LC790722, LC790723, LC790724, LC790725, MH886519, OP871788).

The complete CP sequence of ryegrass mosaic virus (RefSeq: NC_001814) was used as the outgroup.

Initially, sequence alignment was performed using version 1.6 of CLUSTAL W [49] as the DNA weight matrix. To construct the Maximum Likelihood (ML) phylogenetic tree [50,51] the best fitting DNA substitution model had to be determined, choosing the one with the lowest Bayesian Information Criterion (BIC) score [52]. Tree reliability was tested using the bootstrap method with 1,000 pseudo-replicates [53]. The resulting phylogenetic tree was rooted in the outgroup. Additionally, the acronyms of the amino acid triplets responsible for successful vector transmission have also been included after the scientific name of the host.

## Identification of recombination

For recombination analyses, nucleotide sequences were trimmed – when necessary – to start with the conserved GNNSGQ amino acid motif of the nuclear inclusion b (NIb) protein at the 5′ end [41,54,55]. This motif is present in all members of the *Potyvirus*, *Macluravirus*, and *Rymovirus* genera within the *Potyviridae* family. The sequences also included the CP region and the 3′ untranslated region (3′UTR), when available. Alignments were constructed using sequences obtained in this study, along with the following reference sequences from NCBI GenBank: lily mottle virus (AB053256, AB078007, AB090385, AF531458, EU267778, EU348826, FJ618539, JN127335, JN127341, JN848600, JQ361099, KF417755, KF553658, KJ561805, MF781080, MF983709, MH360239, MK368784, MK368788, MK368790, MK368792, MK368793, MK368794, MK368801, MK368802, MK368803, MK368804, MK368806, MK368809, OM311163), ryegrass mosaic virus (NC_001814), snowdrop virus Y (EU927399, LC757029, LC790722, LC790723, LC790724, LC790725, MH886519, OP871788), tobacco etch virus (NC_001555).

The initial alignment was performed in MEGA X [48] using version 1.6 of CLUSTAL W [49] as DNA weight matrix. The best-fitting DNA substitution model was identified by selecting the one with the lowest Bayesian Information Criterion (BIC) score [52]. The resulting data were used to optimize the settings of recombination prediction.

To infer potential recombination events, version 4.101 of the Recombination Detection Program (RDP) was utilized [56,57]. Default settings were used in most cases. Sequences were set as linear and automatic masking was enabled. The highest acceptable *p*-value was set to 0.05 using Bonferroni correction [58]. An automatic exploratory analysis was carried out with the following statistical methods: RDP [59], Chimaera [60], Bootscan/Recscan [61], 3SEQ [62],

GENECONV [63], MaxChi [64], and SiScan [65]. For RDP, reference sequence selection was restricted to internal references. Bootscan analyses used the Jin and Nei model [66]. Both MaxChi and Chimaera were run with variable window sizes, and the fraction of variable sites per window was set to 0.1. SiScan was performed with 1,000 permutations. PhylPro [67] was conFigd to exclude self-comparisons. LARD [68] employed the HKY model [69] with a moving partition scan, while TOPAL [70] used the Jin and Nei model [66] with a transition:transversion ratio of 2. Distance plots and Neighbor-Joining (NJ) trees [71] were also based on the Jin and Nei model [66], using 100 bootstrap replicates [53] and a random seed value of 3. SCHEMA [72] was run with an interaction distance of 4, temperature set to 20, and sequences specified as RNA. Following the exploratory analysis, all predicted recombination events were re-checked using all available methods. Each event was then manually inspected to confirm topological differences between trees, verify breakpoint consistency, and correct discrepancies if necessary. Recombination events supported by fewer than three detection methods were rejected.

## Results

### Serological and molecular identification

Among the ten randomly collected wild *Galanthus nivalis* samples and one cultivated sample, only two tested positive in the ELISA assays. Two specific PCR products were successfully amplified using genus-specific primers for potyviruses, whereas no amplification was detected for *Nepovirus* subgroup A or tomato spotted wilt virus (TSWV). A summary of the serological and molecular testing results is presented in Table 2.

Nucleotide sequence determination and subsequent BLAST analysis of the isolates confirmed that all samples were infected with *Potyvirus lilimaculae* (lily mottle virus, LMoV), based on the species demarcation criteria established in the latest ICTV Taxonomy Profile [46]. A summary of these results and the highest nucleotide identity to GenBank accessions is provided in Table 3.

### Phylogenetic analyses

The best-fitting DNA substitution model was determined to be the HKY model [69], with gamma distribution and invariant sites, which yielded the lowest Bayesian Information Criterion (BIC) value of 9668.851. Phylogenetic analysis revealed that both SVY and LMoV have monophyletic origins and are located on separate branches of the tree with 100% bootstrap support (Fig 1). The LMoV clade further subdivided into two well-defined subclades, consistent with previous reports [73,74].

**Table 2. Summary of the results of the ELISA and PCR tests of snowdrop samples.**

| Sample ID | Potyvirus group ELISA absorbance | | specific PCR product present | | |
|---|---|---|---|---|---|
| | sample | negative control | Nepovirus A | TSWV | Potyvirus |
| 647 | 0.011; 0.015 | 0.006; 0.008 | no | no | no |
| 659 | 0.009; 0.012 | 0.006; 0.008 | no | no | no |
| 666 | 0.007; 0.009 | 0.006; 0.008 | no | no | no |
| 685 | 0.018; 0.022 | 0.004; 0.006 | no | no | yes, two isolates |
| 785 | 0.001; 0.003 | 0.006; 0.008 | no | no | no |
| 804 | 0.001; 0.003 | 0.006; 0.008 | no | no | no |
| 857 | 0.008; 0.009 | 0.006; 0.008 | no | no | no |
| 862 | 0.011; 0.029 | 0.045; 0.053 | no | no | no |
| 863 | 0.027; 0.031 | 0.045; 0.053 | no | no | no |
| 865 | 0.074; 0.079 | 0.021; 0.022 | no | no | yes |
| 885 | 0.012; 0.019 | 0.045; 0.053 | no | no | no |

Table 3. Summary of the results of nucleotide sequence determination and BLAST analysis of the CP regions.

| Sample ID | isolate ID | Identified virus | GenBank acc. nr. | Accession with the highest identity in the CP region | Identity | Identity with the reference sequence of LMoV (NC_005288) |
|---|---|---|---|---|---|---|
| **685** | 685−1 | *Potyvirus lilimaculae* | PV090995 | JF714974 | 96.51% | 92.61% |
| | 685−2 | *Potyvirus lilimaculae* | PV090996 | JF714974 | 96.28% | 91.16% |
| **865** | 865−1 | *Potyvirus lilimaculae* | PV090997 | JF714974 | 96.51% | 92.61% |

Lily mottle virus isolate Handan (JF14974) is from *Narcissu tazetta* var. *chinensis* from Hubei province from China (unpublished).

All LMoV isolates originated from snowdrops were clustered within subgroup II of LMoV. Their closest relatives were LMoV isolates from lilies, notably *Lilium longiflorum* from Australia (JN127341) and *Lilium davidii* var. *unicolor* from China (MF781080). These snowdrop-derived isolates were distantly related to other Hungarian LMoV isolates previously identified from tulips [74]. Interestingly, all LMoV isolates – regardless of their subgroup classification – shared the NAG amino acid triplet associated with vector transmission [75,76]. In contrast, SVY isolates only had DAG triplet. This may indicate that mutation of aspartic acid (D) to asparagine (N) has no or negligible effect of the ability of the pathogen to be effectively transmitted between hosts by vectors.

## Inference of recombination

The Recombination Detection Program did not identify any recombination signals in SVY. However, six potential recombination events were detected in LMoV, of which three were predicted as recombinant with at least three different statistical methods, thereby meeting the arbitrary threshold for reliable detection. All of the predicted events were cases of intraspecific recombination. The program used statistical methods to simulate the statistically most likely parental sequence from the alignment; these accessions are given in parenthesis (Table 4).

No nucleotide sequences were predicted to result from multiple recombination events. In all cases, either the major or the minor parental sequence could not be identified. Most of the predicted recombination breakpoints were located within the NIb and CP regions, with only a single breakpoint detected in the 3′UTR (Fig 2).

In all predicted events, the Chinese lily-derived LMoV isolate OM311163 was predicted as the major parent or used to simulate the major parental sequence in recombination events. Notably, all LMoV isolates from snowdrops were predicted to be recombinant. Isolates 685−1 and 685−2 – both identified from the same plant – shared the initial breakpoint at the start of the conserved GNNSGQ motif in the NIb region. Additionally, isolates 685−2 and 865−1 have the ending breakpoints after the NAG triplet in the CP. Both the GNNSGQ motif and the NAG triplet are highly conserved in LMoV.

The first potential recombination event predicted by the program involved accession JN127341. This isolate had already been reported as a recombinant derived from multiple recombination events [74]. In the earlier study, the initial recombination breakpoint was identified at nt position 645, just prior to the NAG triplet, while the terminal breakpoint was located at nt 1398, corresponding to the carboxyl-terminal region of the CP.

In contrast, the present analysis predicted the first breakpoint at alignment position 19, the last nucleotide at GNNSGQ motif. The ending breakpoint was at nt 744, corresponding to the final nucleotide of the alanine (A) in the NAG triplet (S1 Fig). The predicted major parent was isolate OM311163 – a subgroup II lily isolate from China – while the minor parental sequence was simulated using MK368802, a Hungarian isolate from tulip from subgroup I. Notably, JN127341 locates a distinct branch in both the major and minor parental phylogenies, supporting its unique recombinant origin, as previously proposed.

All LMoV isolates originating from snowdrops were predicted to be recombinant. In each case, the major parent was identified as isolate OM311163 from subgroup II, a lily isolate from China. The minor parent was simulated using MK368801, a Hungarian isolate from tulip from subgroup I.

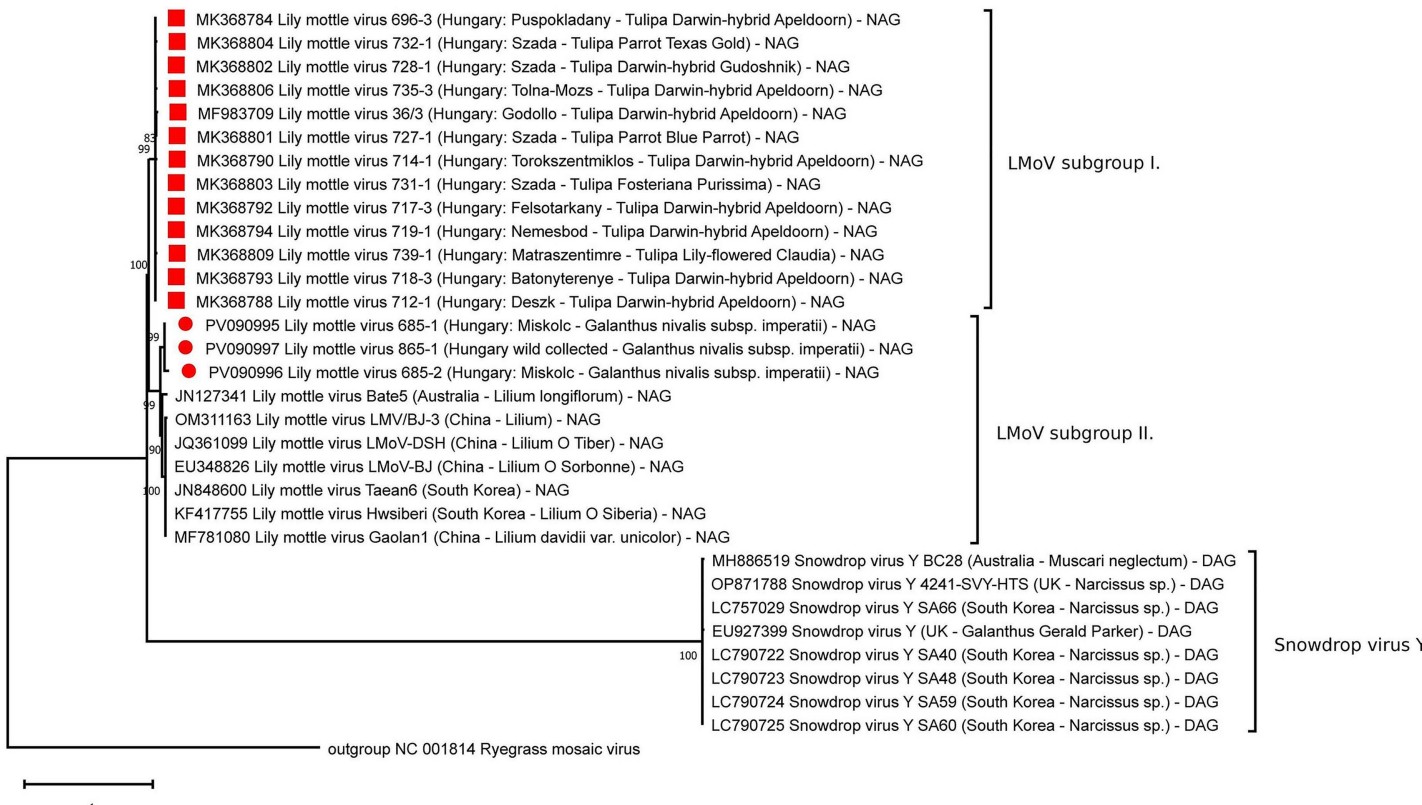

**Fig 1. ML phylogenetic tree of CP sequences built with HKY nucleotide substitution model with gamma distribution, invariant sites and tested with the bootstrap method with 1,000 pseudo-replicates.** Outgroup is ryegrass mosaic virus (RefSeq: NC_001814). Bootstrap values are indicated on the branches as percentage. Red squares indicating isolates originated from Hungary, red dots indicate Hungarian isolates from snowdrops reported in this study.

**Table 4. Summary of predicted and accepted recombination events.**

| Recombination event nr. | Inferred recombinant sequence | Predicted major parent | Predicted minor parent |
|---|---|---|---|
| 1 | JN127341 | OM311163 | unknown (MK368802) |
| 2 | PV090995 PV090996 PV090997 | OM311163 | unknown (MK368801) |
| 3 | AJ310203 | unknown (OM311163) | MF983709 |

The average *p*-values (Bonferroni corrected) were calculated for each recombination event, the highest acceptable *p*-value was 0.05 (S1 Table).

For isolate 685−1 (acc. nr.: PV090995), the first recombination breakpoint was predicted at alignment position 13, which corresponds to the final nucleotide of the serine (S) within the conserved GNNSGQ motif. The ending breakpoint was located at position 698, the first nucleotide of the asparagine (N) within the conserved NAG triplet (Fig 3).

For isolate 685−2 (acc. nr.: PV090996), the first recombination breakpoint was predicted at alignment position 16, corresponding to the second nucleotide of the glutamine (Q) in the conserved GNNSGQ motif. The final breakpoint was located at position 795, the second nucleotide of leucine (L) at amino acid (aa) position 18 of the CP, following the NAG triplet.

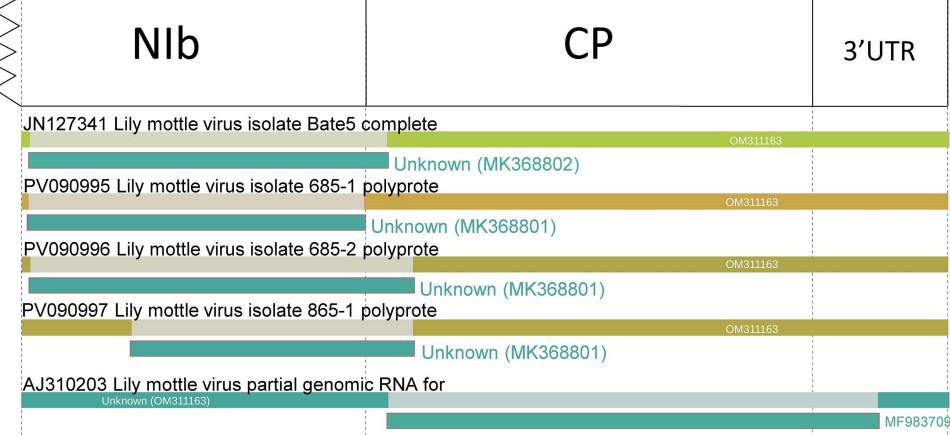

**Fig 2. Schematic display of predicted recombination events on the analyzed region of potyvirus genome map.** First dashed line indicates the beginning of the fragment (GNNSGQ motif) used. Second dashed line indicates the NIb/CP cleavage site, the third line the stop codon of the polyprotein and the last represents the end of the 3′UTR preceding the polyA tail.

In isolate 865−1 (acc. nr.: PV090997), the first breakpoint was predicted at alignment position 224. This site corresponds to a gap in the alignment, as this isolate was obtained via direct sequencing and was shorter than other snowdrop-derived sequences. The final breakpoint was again predicted at position 795, the second nucleotide of leucine (L) at aa position 18 of the CP, after the NAG triplet, just like in isolate 685−2.

All three snowdrop-derived LMoV isolates formed a distinct subclade in both the major and minor parental phylogenetic trees, indicating a strong possibility of recombination events in their ancestry and distinct origin from both parents. Similar to Bate5 isolate (JN127341), their definite separation from both subgroups I and II implies that these snowdrop isolates may have a more complex evolutionary history, potentially involving multiple recombination events that remain to be identified.

The third recombination event was predicted in accession AJ310203, a lily-derived LMoV isolate from China. The program used OM311163 from subgroup II – a lily isolate from China – to simulate the major parent. The predicted minor parent was MF963709, a Hungarian tulip isolate from subgroup I, in accordance with previous findings [74]. The first recombination breakpoint was predicted at alignment position 744, corresponding to the last nt position of the alanine (A) in the NAG triplet. The same recombination breakpoint was identified in accession JN127341 as the ending breakpoint. The final breakpoint was predicted at position 1739, within the 3′UTR (S2 Fig).

In both parental trees, AJ310203 is located on a distinct branch. However, in each case, the sequence segment corresponding to the major parent falls within subgroup II, while the segment corresponding to the minor parent clusters within subgroup I. This pattern supports the hypothesis that AJ310203 resulted from a single, well-defined recombination event.

### Fulfillment of Koch's postulates

The results of the sap inoculation experiments were evaluated using ELISA and RT-PCR in the next growing season following inoculation. Snowdrops, like the majority of non-tropical geophytes, exhibit a distinct seasonal growth rhythm. During the emergence and flowering phases, no mitotic activity occurs in the leaves, pedicels, or tepals; instead, existing cells undergo expansion [47,77]. Mitotic activity is confined to the bulb, where new buds, leaves, floral structures, and scales are formed. In *Galanthus*, this developmental activity begins while the plant is still flowering and continues into the summer months [78]. Consequently, visual symptoms of the infection in non-tropical geophytes usually appear only in the subsequent growing season, when newly formed leaves and flowers emerge above ground. Both 685 and 865 samples

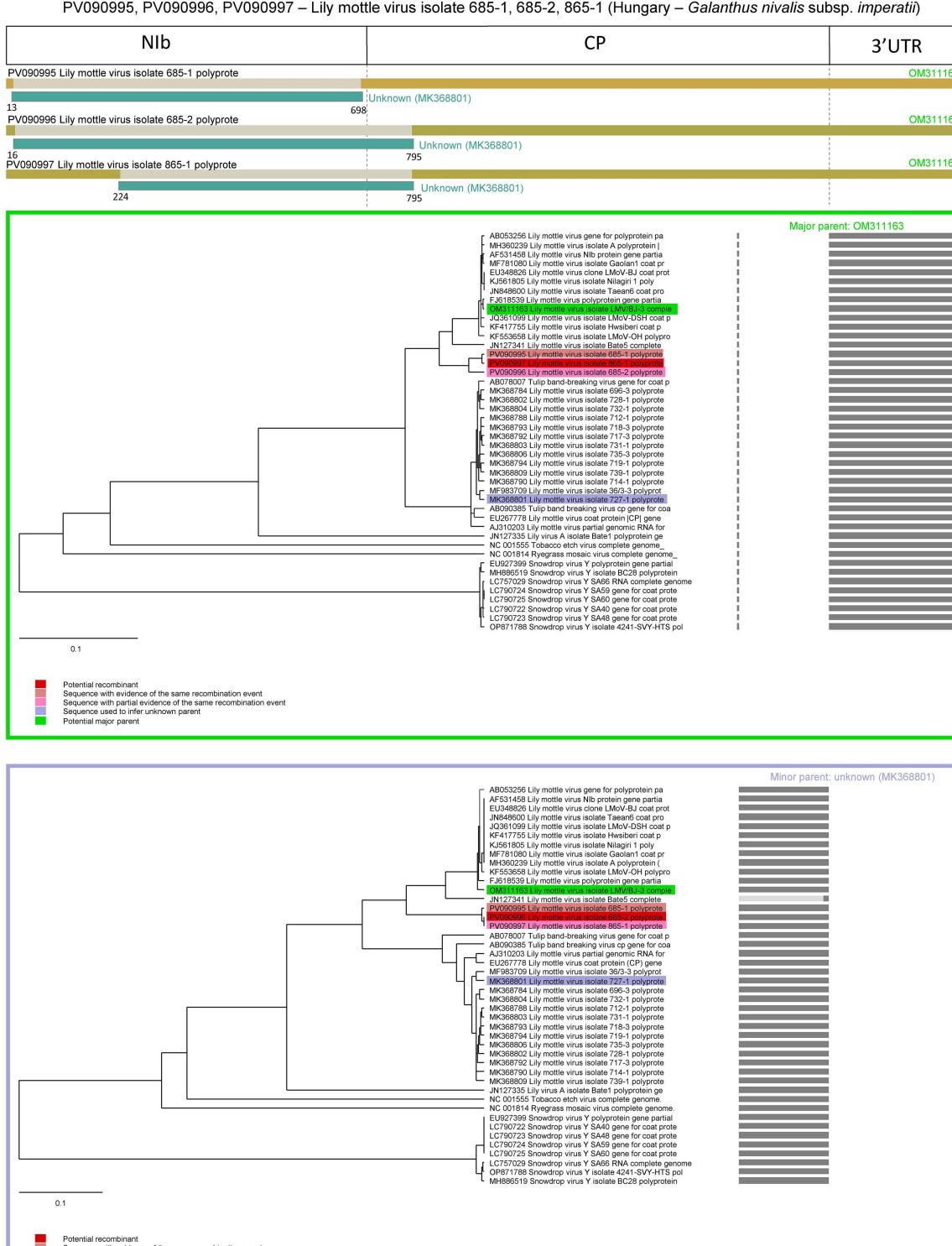

**Fig 3. Schematic display of the second recombination event.** Neighbour Joining (NJ) tree of inferred major parent is on top, minor parent is on the bottom. Bars to the right of the trees represent the sequence regions from which the trees were built. Red background indicates recombinant sequence, the lilac background indicates inferred minor parent and light green major parent. Predicted recombination breakpoints are given in nucleotide base numbers in the alignment..

were sap-inoculated onto *Galanthus elwesii* plants. In both cases, only the sap-inoculated plants tested positive by ELISA (Table 5), confirming successful transmission.

Similarly, only the sap-inoculated plants yielded specific PCR products when tested with the poty7941 and polyT$_2$ primers. Results of the sequencing revealed 100% sequence identity of the sap-inoculated plants with the original samples. Notably, both the original samples and the sap-inoculated plants remained asymptomatic throughout the entire observation period (Fig 4). Compared to the healthy plants we have not observed obvious reduction of plant size or flower production by visual assessment. Asymptomatic virus infection of wild plants have been reported earlier, mostly from weeds [79–81].

## Discussion

Virus research have long been driven by the analysis of symptomatic crop plants, which have led to identification of some mild or low symptom inducing viruses: lily symptomless virus, saffron latent virus, shallot latent virus, wild onion symptomless virus, eggplant mosaic virus infection on *Abelia grandiflora*, tobacco necrosis virus on *Anemone coronaria*, Colombian datura virus on *Brugmansia*, just to name a few [36].

**Table 5. Summary of the ELISA test of the mock and sap inoculated plants.**

| Sample | Potyvirus group ELISA absorbance | |
|---|---|---|
| | sample | negative control |
| mock inoculated 1 | 0.009; 0.012 | 0.004; 0.007 |
| mock inoculated 2 | 0.007; 0.009 | 0.004; 0.007 |
| sap inoculated with 685 | 0.045; 0.048 | 0.004; 0.007 |
| sap inoculated with 865 | 0.035; 0.037 | 0.004; 0.007 |

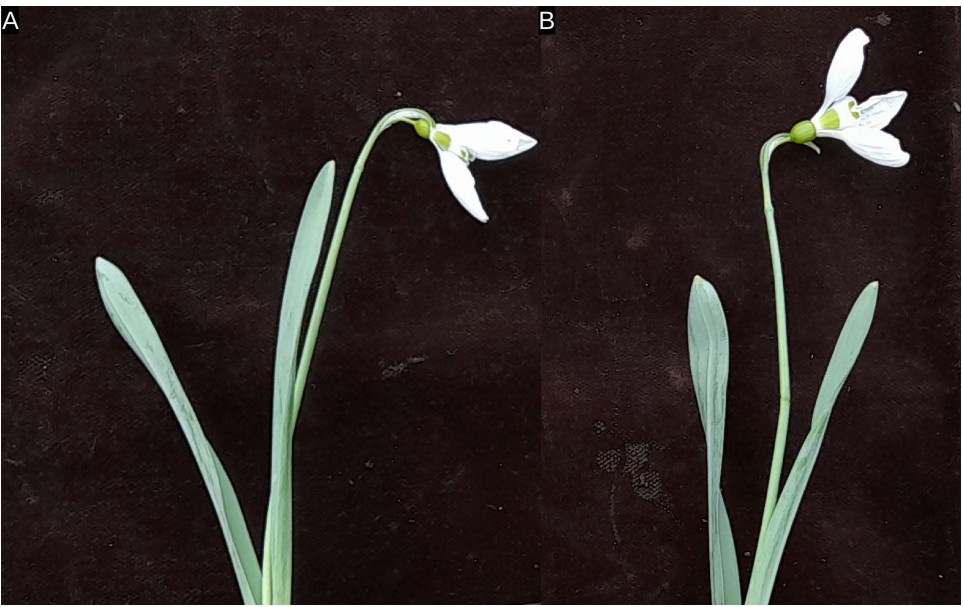

**Fig 4. Results of the inoculation assay.** On the left (A) is the mock-inoculated control, on the right (B) is the plant inoculated with the sap of sample 685. The photo was taken one year after the inoculation.

With the advancement of sequencing there have been several reports of plants that can harbor viruses asymptomatically [79,80,82–91], but these reports usually target economically important crops, not wild plants [80]. Weeds are known reservoirs of economically important crop viruses [92,93], but non-weed wild plants can also harbor multiple pathogens, and can be co-infected by multiple species and strains of viruses, while remaining asymptomatic [79,81,94,95]. Understanding long term host-virus interactions in wild populations of plants may become more important as environmental and cultivation conditions change [80,94,96]. Viruses may jump to new hosts, trigger epidemics, and persist in perennial plants as long-term reservoirs [97]. Vegetative propagation of host plants can positively affect the nucleotide diversity of viruses, but also impose stronger genetic bottlenecks, which influence virus evolution and affect recombination [94,98].

Snowdrops are significant in the horticultural trade [6,7], valued not only for their ornamental appeal but also for their medicinal properties [13–15]. However, natural populations are vulnerable to decline due to over-collection, poaching, and habitat loss. To mitigate these threats and regulate the global trade and wild collection of snowdrops, all species within the *Galanthus* genus are listed under CITES protection [99]. In addition, both the genus and its natural habitats are protected under European Union legislation [100,101] and Hungarian national law [23,24].

Despite their ecological and economic importance, virus research on *Galanthus* species and cultivars remains limited. To date, only a single virus has been described from *Galanthus*: Snowdrop virus Y (SVY), a tentative member of the genus *Potyvirus*, family *Potyviridae*.

To our knowledge, lily mottle virus (LMoV) has previously been reported only from symptomatic plants of *Alstroemeria* [102], *Cichorium endivia* subsp. *latifolium* [103], *Lilium* [36,104–109], *Narcissus* [36], *Tulipa* [36,74,105,110–114], and *Zantedeschia* [102]. This study would represent the first report of *Galanthus* as an asymptomatic host of LMoV, while additional LMoV sequences are available in GenBank from *Gladiolus*, *Hyacinthus*, and *Muscari*, these hosts have not been formally reported in peer-reviewed publications. Nevertheless, the diversity of reported hosts suggests that LMoV may have a generalist strategy and preference for monocotyledonous geophyte plants.

DAG triplet in the CP region of potyviruses plays a pivotal role in vector transmission [115,116]. Interestingly, all LMoV isolates examined in this study contain the NAG triplet, regardless of whether they belong to subgroup I or II, while SVY isolates retain the DAG triplet. This amino acid mutation has been previously reported in bean yellow mosaic virus (BYMV) and clover yellow vein virus, and it appears that it is specific to the BYMV subgroup in genus *Potyvirus* [75]. This may indicate that mutation of aspartic acid (D) to asparagine (N) has no or negligible effect of the ability of these pathogens to be effectively transmitted between hosts by vectors, or that there is also a mutation in the Helper-component proteinase (HC-Pro) region and mutation of aspartic acid to asparagine was necessary or beneficial for vector transmissibility [117].

Recombination plays a significant role in virus evolution and potyviruses are known for their high recombination rates [118–121]. All snowdrop isolates of LMoV showed very high probability to be the result of recombination, which was substantiated by strong recombination signals with eight different statistical methods. Because of their separation to a distinct subclade of subgroup II and their close relatedness to LMoV isolate Baste5 (JN127341) – which was earlier reported to have multiple recombination events in its ancestry [74] – it can be presumed, that all snowdrop isolates may have been resulted of multiple recombination events, but this has to be proven by further analysis.

Notably, recombination was inferred in subgroup II sequences [73,74], while not in subgroup I. This raises the possibility that subgroup II may consist entirely of recombinant lineages. If substantiated by additional evidence, this could warrant the reclassification of subgroup II as "recombinant subgroup".

Recombination cold- and hotspots have been identified in several virus species [122,123]. In the case of the LMoV snowdrop isolates, all predicted recombination breakpoints – excluding those falling in alignment gaps – were located in conserved genomic regions. Specifically, the initial breakpoints were consistently found in or near the GNNSGQ motif within the NIb region, a conserved domain shared across the *Potyvirus*, *Macluravirus*, and *Rymovirus* genera [41,54,55]. The corresponding final breakpoints were located in or near the NAG triplet, part of the conserved ANETLNAG motif at

the amino-terminal end of the coat protein. In the case of the LMoV Bate5 isolate (JN127341) the predicted breakpoints were in the same conserved domains as for the snowdrop isolates. However, in the case of Hangzhou isolate (AJ310203) the predicted breakpoints were different. The initial breakpoint was in the NAG triplet, while the final breakpoint was in the 3'UTR. These results align with previous findings: the beginning breakpoint of the first event and the ending breakpoint of the fifth event in Bate5 isolate and the initial breakpoint of the fourth event in Hangzhou isolate was predicted in the ANETLNAG motif, while the final breakpoint of the latter isolate was predicted in the 3'UTR [74]. These conserved regions provide important structural and functional domains in peptides and provides 3D stability of the RNA molecule. Their recurrent involvement in recombination events suggests that such conserved regions may also serve as recombination hotspots in LMoV—and potentially more broadly across the *Potyviridae* family.

The detection of an economically important virus such as LMoV in an asymptomatic wild plant within a protected Natura2000 site holds significant implications for nature conservation, horticulture, phytosanitary practices, and our understanding of *Potyviridae* evolution.

The asymptomatic nature of the infection complicates visual detection, making it impossible to assess the virus's prevalence or its potential impact on wild snowdrop populations without molecular testing. This underscores the need for wide-scale sampling and long-term monitoring of both host and pathogen in natural habitats.

LMoV is readily transmitted by plant sap. Therefore, human activities such as flower picking, plant digging, mowing, or trampling can create wounds that facilitate viral transmission. These adverse human effects are partially eliminated by declaring snowdrops and their habitat protected in Hungary [23]. Due to the sap-transmissibility of LMoV, it is advisable to avoid cutting snowdrop foliage or mowing their habitats during the vegetative phase to minimize the risk of virus spread.

Currently, it is unknown whether LMoV can be transmitted through seed in *Galanthus*. Until this question is resolved, it is recommended that only locally harvested seeds and bulbs be used for *in situ* conservation efforts and population restoration projects to minimize the risk of spreading or introducing infected plants to a healthy population.

Tulips and lilies are the two most economically significant hosts of LMoV in Europe. In the Netherlands alone, the total area of lily cultivation in 2024 reached 4,251.8 hectares, while tulips were cultivated on 13,175.15 hectares [124,125]. Both crops are subject to stringent phytosanitary regulations. For instance, propagation material for lilies must be entirely free of LMoV infection, and commercial bulbs must have an infection rate below 5% within any given lot. Mandatory laboratory testing and field inspections are required for both crops, and any lot that fails to meet these standards must be destroyed [126,127]. Given that snowdrops can serve as asymptomatic carriers of LMoV, it would be prudent to establish a minimum isolation distance of at least 300 meters between snowdrop populations and commercial lily or tulip cultivation areas to reduce the risk of virus transmission. Moreover, the identity of the vector(s) responsible for transmitting LMoV between snowdrops or from snowdrops to other hosts remains unknown, highlighting the importance for further research into the epidemiology of this virus.

A latent infection typically indicates a balanced host-pathogen relationship, suggesting that the two have co-evolved over an extended period of time [79]. Such host-pathogen dynamics have been extensively studied in humans and animals [128–130], as well as in economically important crops [82–84,87,131–135]. The asymptomatic nature of LMoV infection in *Galanthus* may imply that *Galanthus*, or one of its close relatives – such as *Acis*, *Leucojum*, *Narcissus*, *Pancratium*, *Sternbergia*, or *Vagaria* – could be among the original natural hosts in which lily mottle virus evolved. In this scenario, the symptomatic infections observed in other plant species may reflect an ongoing process of viral adaptation to new, less compatible hosts.

If this hypothesis holds true, it would suggest that LMoV is native to Hungary, similarly to meadow saffron breaking virus [136], and that lilies and other affected hosts represent secondary spillover hosts. This evolutionary origin could also help explain the existence of the two distinct phylogenetic subgroups observed within LMoV.

Earlier studies estimated that the common ancestor of potyviruses may have diverged approximately 6,000–7,000 years ago – coinciding with the advent of agriculture and the exchange of propagating materials by early human societies

[137]. More recent analyses suggest an earlier divergence, placing the initial radiation of potyviruses between 15,000 and 30,000 years ago, likely originating from an ancestral Eurasian grass host [138].

Given the conservation status of the *Galanthus* genus, any further research – particularly large-scale sampling efforts – requires official permission from the appropriate governmental authorities. Securing these permits, along with conducting comprehensive studies, would necessitate significant financial resources.

## Supporting information

**S1 Fig. Schematic display of the first recombination event.** Neighbour Joining (NJ) tree of inferred major parent is on top, minor parent is on the bottom. Bars to the right of the trees represent the nucleotide sequence regions from which the trees were built. Red background indicates recombinant sequence, the lilac background indicates inferred minor parent and light green major parent. Predicted recombination breakpoints are given in nucleotide base numbers in the alignment. (JPG)

**S2 Fig. Schematic display of the third recombination event.** Neighbour Joining (NJ) tree of inferred major parent is on top, minor parent is on the bottom. Bars to the right of the trees represent the sequence regions from which the trees were built. Red background indicates recombinant sequence, the blue background indicates minor parent and light green inferred major parent. Predicted recombination breakpoints are given in nucleotide base numbers in the alignment. (JPG)

**S1 Table. Summary of confirmation tables for the accepted recombination events.** Highest acceptable *p*-value was 0.05.
(DOCX)

## Acknowledgments

We would like to express our gratitude towards Joe Sharman – a renowned plantsman, "Galanthophile" and owner of Monksilver Nursery, UK – who helped to identify *G. nivalis* subsp. *imperatii* in 2004.

**Declaration of generative AI and AI-assisted technologies in the writing process**

During the preparation of this work the authors used OpenAI ChatGPT-4 in order to check grammar, spelling, and improve readability and language. After using this tool, the authors reviewed and edited the content as needed and take full responsibility for the content of the published article.

## Author contributions

**Conceptualization:** János Ágoston, Asztéria Almási, Katalin Salánki, László Palkovics.

**Data curation:** János Ágoston.

**Formal analysis:** János Ágoston.

**Funding acquisition:** Katalin Salánki, László Palkovics.

**Investigation:** János Ágoston.

**Methodology:** János Ágoston, Asztéria Almási, Katalin Salánki, László Palkovics.

**Project administration:** János Ágoston, Katalin Salánki, László Palkovics.

**Resources:** János Ágoston, Katalin Salánki, László Palkovics.

**Supervision:** Asztéria Almási, Katalin Salánki, László Palkovics.

**Visualization:** János Ágoston.

**Writing – original draft:** János Ágoston, Asztéria Almási, Katalin Salánki, László Palkovics.

**Writing – review & editing:** János Ágoston, Asztéria Almási, Katalin Salánki, László Palkovics.

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
