## [Decision Letter · Decision Letter 0]

22 Oct 2025

Dear Dr. Palkovics,

Thank you for submitting your manuscript to PLOS ONE. After careful consideration, we feel that it has merit but does not fully meet PLOS ONE’s publication criteria as it currently stands. Therefore, we invite you to submit a revised version of the manuscript that addresses the points raised during the review process.

We look forward to receiving your revised manuscript.

Kind regards,

Adedapo Olutola Adediji, Ph.D.

Academic Editor

PLOS ONE

Journal Requirements:

“HUN-REN Hungarian Research Network (project number: 3200107).

JA’s research was partly funded by the North American Lily Society Research Trust Fund Grant (grant number: 2024-02).”

4. Please amend your authorship list in your manuscript file to include author László Palkovics.

Additional Editor Comments (if provided):

The paper describes the occurrence of potyvirus in asymptomatic Galanthus nivalis. However, the reviewers provided critical and very sacrosanct grounds for the paper to be rewritten and additional data provided. I will recommend a major revision. In addition, there are major grammatical improvements to be made to the paper. In addition, there is a high similarity match with a paper previously published by the authors: this must be corrected before any consideration is given to publish or not.

Reviewers' comments:

Reviewer's Responses to Questions

**Comments to the Author**

1. Is the manuscript technically sound, and do the data support the conclusions?

Reviewer #1: Yes

Reviewer #2: Yes

Reviewer #3: No

2. Has the statistical analysis been performed appropriately and rigorously?

Reviewer #1: N/A

Reviewer #2: Yes

Reviewer #3: Yes

3. Have the authors made all data underlying the findings in their manuscript fully available?

Reviewer #1: Yes

Reviewer #2: Yes

Reviewer #3: Yes

4. Is the manuscript presented in an intelligible fashion and written in standard English?

Reviewer #1: No

Reviewer #2: Yes

Reviewer #3: Yes

Reviewer #1: The manuscript “Recombinant Potyvirus lilimaculae in asymptomatic Galanthus nivalis: Ecological and evolutionary implications” presents the results of viruses survey in asymptomatic snowdrops plants mainly from wild protected areas of Hungary. The authors found a virus species already described in other hosts, the lily mottle virus (Potyvirus lilimaculae). They also performed molecular phylogenetic studies based on CP nucleotide sequences and recombination analysis based on partial NIb and CP nucleotide sequences.

The research was well conducted and results are important in relation to biodiversity aspect, although some aspects from the agronomic perspective could be better explored. I feel the conclusions by authors sometimes were overstated, mainly because the study was conducted with partial sequences. I also suggest the authors to conduct a phylogenetic analysis with predicted CP amino acids sequences. Some tables and figures can be moved to supplementary file, since are not essential for understanding by readers (I commented in the attached file). Other specific comments and suggestions are pointed out in the attached file.

The manuscipt needs an english language revision.

Reviewer #2: I have critically reviewed the manuscript title “Recombinant Potyvirus lilimaculae in asymptomatic Galanthus nivalis: Ecological and evolutionary implications”, which describes the asymptomatic infection of recombinant isolates of Lily mottle virus in Galanthus species. The authors used serological as well as molecular approaches for the detection of potyvirus group. There used mechanical inoculation method and found positive infection in the inoculated plants during the following year. Recombination analyses revealed the recombination events in the all three isolates studied. The study describes well the importance and novelty of the research in wild plants which act as reservoir hosts for viruses asymptomatically. Methodology section have been explained well to reproduce the experimental results. Results and discussion are well supported with the literature about recent studies related to potyviruses in wild plants.

I would recommend the acceptance of manuscript for publication in its current form.

Reviewer #3: This manuscript reports a possible recombinant isolate of LMoV. However, this document has multiple errors and lack of information. Among the main issues identified: The recombination analysis is based entirely on a small fragment of the viral genome (1689 bp), rather than the entire genome. The three identified LMoV isolates have >96% similarity to the same accession deposited in GenBank. The mechanical transmission experiment (bio-indexing) only demonstrated the infectivity of the virus in the inoculated plants, but no pathogenicity, no symptoms or damage were ever observed. It is well known that potiviruses, including LMoV, have a high degree of recombination, so what would be the originality of this analysis in this work? To say that Galanthus could be one of the natural hosts of LMoV and that this virus could also be native to Hungary, is very risky, and no information or data that could support such claims is presented. Finally, I suggest restructuring this work and presenting it as a first report of LMoV infecting Galanthus in Hungary.

**Do you want your identity to be public for this peer review?** For information about this choice, including consent withdrawal, please see our Privacy Policy

Reviewer #1: No

Reviewer #2: **Yes:** Dr. Khadim Hussain

Reviewer #3: No

---

## [Author Response · Author response to Decision Letter 1]

7 Nov 2025

Response to Editors and Reviewers’ Comments

Manuscript ID: PONE-D-25-31948

Title: Recombinant Potyvirus lilimaculae in asymptomatic Galanthus nivalis: Ecological and evolutionary implications

Dear Editor,

We thank the Editor and Reviewers for the effort and time put into the review of our manuscript (MS).

We have carefully considered the comments and we provide responses to them. We gave line numbering as is in the updated and corrected MS.

Comments from Reviewer #1:

1. Several edits in the text

Answer:

We are thankful for the reviewer for his/her edits in the text, which are implemented in the final MS. We also have shortened the Introduction part as advised.

2. “You can put this information as a disclosure in the end of the manuscript”

Answer:

Our permit conditions require us to indicate the permit numbers in the materials and methods section and the journal requires us to indicate the permits in the ethical statements section. To conform to both requirements the permit numbers are stated at both places.

3. “Please give a more informative title”

Answer:

Corrected.

4. “What is the PCR reagent?”

Answer:

This information is now added to the MS

5. “This information can be removed, since it is not essential”

Answer:

Our permit conditions are very strict and require us to be fully transparent. We must publish all permit related plant/seed collection, growing, germination information, including the failures over our regular research. Noncompliance with the permit conditions will result in banning of the permittee from all future applications, furthermore the permittee will receive fines and may be subjected to administrative lawsuits by the Government Office and the National Parks. This is the reason why we would like to keep this section in the MS.

6. line 241: “Please give a more informative title”

Answer:

Corrected.

7. line 268: “Is there a previous report about this in LMoV?”

Answer:

To our knowledge the NAG triplet responsible for aphid transmission in Potyvirus genus was researched by Shukla [1], then later by Uyeda [2]. Both reported that NAG triplets (instead of DAG) are specific for BYMV subgroup in genus Potyvirus, and LMoV is a member of this subgroup. Mutation of the DAG triplet to NAG has also been reported by Nigam [3] and Gadhave [4]. Both reported that there was no loss of aphid transmissibility.

8. line 257: “Please give a more informative title”

Answer:

We believe that this title contains all the information related to the table. We have no idea for a more informative title.

9. line 282: “In my opinion this table can be moved to supplementary material , it is not essential for readers”

Answer:

Agreed, that table is now Supplementary Table 1. The rest of the tables are renumbered in order.

10. line 311: “Suggestion: put the fig 3 and fig 5 as suppl. Material. The fig 4 with the isolates from this study can remain in the main text.”

Answer:

Agreed, Fig 3 is now Supplementary Figure 1. Fig 5 is now Supplementary Figure 2. The rest of the figures are renumbered in order.

11. line 373: “The absorbance values are low in field and sap transmitted samples, please comment this (maybe a low virus concentration)”

Answer:

OD measurements are “low” because each measurement was corrected with the blank as we stated in the Serological Tests subsection of Materials and Methods. These results are still significant, because we considered a sample positive when the corrected OD was 3 times higher than the corrected average OD of the negative controls. In our earlier report [5] LMoV infected tulips have shown even less OD values than in Galanthus.

12. line 378: “Did you observe changes in flower production, plant size, etc?”

Answer:

We have added the following sentence to the MS: Compared to the healthy plants we have not observed obvious reduction of plant size or flower production by visual assessment.

13. line 478: “This sentence need to be more carefully addressed. The sequence is too short and Nib/CP are not the region that best reflect the information of entire genome (CI/ HC-Pro). So it is necessary to perform more sequencing of the entire genome (or at least HC-Pro/CI) to open this possibility. ”

Answer:

We based our hypothesis of LMoV might being native in Hungary to the facts that we have identified LMoV in wild Galanthus nivalis collected at a Natura2000 protected site far away from human interference. We have identified the pathogen with ELISA and RT-PCR methods. The latest species demarcation criteria of the 2022 ICTV Virus Taxonomy Profile [6] allows the identification of a Potyvirid on the complete CP region. The fact that usually asymptomatic infection indicates a long co-evolution of host and pathogen suggests that LMoV might be native. Throughout the paragraph we used conditional sentences, because for the moment these are just hypothesizes and implications, however these are normal part of any Discussion section of a scientific article. We think that Potyvirus evolution is in progress not a done thing and there are several questions unanswered from where LMoV have been originated, but we also think that the results presented in this paper sheds more light on the possible origins of LMoV. This may motivate other researchers to look for LMoV in Galanthus, Acis, Leucojum, Narcissus, Pancratium, Sternbergia, or Vagaria genera to either support or contradict our hypothesis presented in this MS.

Comments from Reviewer #2:

14. Several edits in the text

Answer:

We are thankful for the reviewer for his/her edits in the text, which are implemented in the final MS taking into considerations of the comments of Reviewer #1.

15. line 58: “Delete this parapgraph: it is not relevant and adds only bogous information to the article.”

Answer:

We would like to keep the first part of this paragraph in, because natural and artificial vegetative propagation of a host is important for virus evolution, which is elaborated in the Discussion section. We also used several references in the paragraph, which would indicate that these propagation methods have been practiced and reported by several authors, they are not our imagination. We put this first part to line 85.

16. line 80: “Delete this parapgraph: it is not relevant and adds only bogous information to the article.”

Answer:

We would like to keep this paragraph in. It contains relevant information about the Hungarian snowdrop population’s conservation status and nursery growth rate for possible future re-settlement projects and also possible grant applications for these projects.

17. line 89: “There has to be a logical flow into the subject of viruses within this species. A paragraph linking the cultivation of the plant via vegetative propagation to the putative spread of pathogens might be a good entry. Then, the subject can focus on virus infections.”

Answer:

We have reorganized the Introduction section as advised.

18. line 98: “This is unclear. Is this a Latinized binomial nomenclature for a scientific name or a varietal description? This must be clearly written!”

Answer:

Narcissus is the scientific name of the genus for the daffodil plant, daffodils deleted.

19. line 100: “Delete this parapgraph: it is not relevant and adds only bogous information to the article.”

Answer:

Deleted.

20. line 103: “Remove: this is irrelevant here as this is already part of the introduction in some way.”

Answer:

Deleted.

21. line 118: “Remove this column, since this not providing any different information.”

Answer:

Deleted. However the column on the right have to stay unchanged, because the permit required us to count the number of plants in a tuft. Our strict permit conditions are elaborated in answer #5 for the question of Reviewer #1.

22. line 152: “Remove this, as this is already presenting RESULTS within METHODOLOGY.”

Answer:

Deleted.

23. line 161: “Remove this: these information are not needed.”

This is a common suggestion wit Reviewer #1

Answer:

Our permit conditions are very strict and require us to be fully transparent. We must publish all permit related plant/seed collection, growing, germination information, including the failures over our regular research. Noncompliance with the permit conditions will result in banning of the permittee from all future applications, furthermore the permittee will receive fines and may be subjected to administrative lawsuits by the Government Office and the National Parks. This is the reason why we would like to keep this section in the MS.

24. line 200: “This is very unclear!”

Answer:

We meant the DAG triplets and NAG triplets responsible for aphid transmission of potyvirids. This way, on the phylogenetic tree one can clearly see that SVY only has DAG triplets while LMoV has NAG triplets. Importance of these amino acid triplets are elaborated in the Discussion section.

25. line 237: “There were more than ten wild samples provided, as detailed in Table 1. This must be properly harmonized and corrected to provide the correct information.”

Answer:

We agree with the reviewer. This is now corrected in the MS.

26. line 243: “Two isolates were positively identified in line 293 but here three isolates were mentioned. Why the disparity? This must be cleared.”

Answer:

We have added: “yes, two isolates” to table 2. We also deleted three from the sentence and is later explained in table 3.

References

1. Shukla DD, Frenkel MJ, Ward CW. Structure and function of the potyvirus genome with special reference to the coat protein coding region. Canadian Journal of Plant Pathology. 1991;13: 178–191. doi:10.1080/07060669109500953

2. Uyeda I. Bean yellow mosaic virus subgroup; search for the group specific sequences in the 3′ terminal region of the genome. In: Barnett OW, editor. Potyvirus Taxonomy. Vienna: Springer Vienna; 1992. pp. 377–385. doi:10.1007/978-3-7091-6920-9_40

3. Nigam D, LaTourrette K, Souza PFN, Garcia-Ruiz H. Genome-wide variation in potyviruses. Front Plant Sci. 2019;10: 1439. doi:10.3389/fpls.2019.01439

4. Gadhave KR, Gautam S, Rasmussen DA, Srinivasan R. Aphid transmission of Potyvirus: the largest plant-infecting RNA virus genus. Viruses. 2020;12: 773. doi:10.3390/v12070773

5. Ágoston J, Almási A, Salánki K, Palkovics L. Genetic diversity of potyviruses associated with tulip breaking syndrome. Plants. 2020;9: 1807 (1–25). doi:10.3390/plants9121807

6. Inoue-Nagata AK, Jordan R, Kreuze J, Li F, López-Moya JJ, Mäkinen K, et al. ICTV Virus Taxonomy Profile: Potyviridae 2022. Journal of General Virology. 2022;103. doi:10.1099/jgv.0.001738

Sincerely yours,

László Palkovics

---

## [Decision Letter · Decision Letter 1]

4 Mar 2026

Recombinant Potyvirus lilimaculae in asymptomatic Galanthus nivalis: Ecological and evolutionary implications

PONE-D-25-31948R1

Dear Dr. László Palkovics,

We’re pleased to inform you that your manuscript has been judged scientifically suitable for publication and will be formally accepted for publication once it meets all outstanding technical requirements.

Kind regards,

Cheorl-Ho Kim, Ph.D.

Academic Editor

PLOS One

Additional Editor Comments (optional):

Dear Dr László Palkovics,

Thank you for your appropriate revision of your original manuscript.

I have rechecked the revision as you well did and found it acceptable for publication in Plos One.

I would like to express my appreciation for your submission to Plos One.

Thank you

Sincerely

Cheorl-Ho Kim

Editor

Reviewers' comments:

Reviewer's Responses to Questions

**Comments to the Author**

Reviewer #1: All comments have been addressed

Reviewer #4: (No Response)

2. Is the manuscript technically sound, and do the data support the conclusions?

Reviewer #1: Yes

Reviewer #4: Yes

3. Has the statistical analysis been performed appropriately and rigorously?

Reviewer #1: N/A

Reviewer #4: Yes

4. Have the authors made all data underlying the findings in their manuscript fully available?

Reviewer #1: Yes

Reviewer #4: Yes

5. Is the manuscript presented in an intelligible fashion and written in standard English?

Reviewer #1: Yes

Reviewer #4: Yes

Reviewer #1: The revised version of manuscript "Recombinant Potyvirus lilimaculae in asymptomatic Galanthus nivalis: Ecological and

evolutionary implications" presents significant improvement compared to the first version. Most of the concerns of reviewers were suitably adressed by authors. I have some minor corrections/suggestions presented directly in the attached file.

Reviewer #4: The revised manuscript has improved substantially in structure, clarity, and scientific presentation. Most previous concerns have been adequately addressed.

The study provides novel and valuable data on Potyvirus lilimaculae (LMoV) in asymptomatic Galanthus nivalis, including molecular identification, phylogenetic analysis, recombination inference, and biological validation via sap inoculation. The integration of virology with conservation implications is particularly interesting and relevant.

The manuscript is close to being acceptable. However, a few points require minor revision before final acceptance (refer to the attached file).

**Do you want your identity to be public for this peer review?** For information about this choice, including consent withdrawal, please see our Privacy Policy

Reviewer #1: No

Reviewer #4: **Yes:** Zaiton Sapak

---

## [Editor Report · Acceptance letter]

PONE-D-25-31948R1

PLOS One

Dear Dr. Palkovics,

I'm pleased to inform you that your manuscript has been deemed suitable for publication in PLOS One. Congratulations! Your manuscript is now being handed over to our production team.

Kind regards,

on behalf of

Professor Cheorl-Ho Kim

Academic Editor

PLOS One